# Do Active Commuters Feel More Competent and Vital? A Self-Organizing Maps Analysis in University Students

**DOI:** 10.3390/ijerph19127239

**Published:** 2022-06-13

**Authors:** Joachim Bachner, Xavier García-Massó, Isabel Castillo, Filip Mess, Javier Molina-García

**Affiliations:** 1Associate Professorship of Didactics in Sport and Health, Technical University of Munich, 80992 Munich, Germany; filip.mess@tum.de; 2Department of Teaching of Musical, Visual and Corporal Expression, University of Valencia, 46022 Valencia, Spain; xavier.garcia@uv.es (X.G.-M.); javier.molina@uv.es (J.M.-G.); 3AFIPS Research Group, University of Valencia, 46022 Valencia, Spain; isabel.castillo@uv.es; 4Department of Social Psychology, University of Valencia, 46010 Valencia, Spain

**Keywords:** person-oriented analysis, cluster, profile, well-being, competence, vitality, physical activity, active commuting, university, outdoor

## Abstract

University students represent a population that faces high risks regarding physical inactivity. Research suggests that a regular engagement in physical activity (PA) may be more likely established when it leads to the experience of subjective vitality. Subjective vitality, in turn, is more likely achieved through physical activities that individuals feel competent in, and that take place in natural outdoor environments. An activity that may fulfill these conditions is active commuting to and from university (ACU). To examine whether and in which form ACU can combine this promising pattern of aspects, a person-oriented analysis was conducted. The sample contained 484 university students (59.3% females). Leisure-time PA, ACU by walking, ACU by cycling, subjective vitality, PA-related competence and body mass index were included as input variables in a self-organizing maps analysis. For both female and male university students, the identified clusters indicated that students who intensively engaged in ACU did not exhibit subjective vitality levels above average. Consistently, they did not show elevated levels of PA-related competence, which suggests that ACU does not support the perception of their physical abilities. Considerations regarding urban university environments lacking sufficient natural elements finally add to the conclusion that engaging in ACU does not suffice to establish a vitality-supportive and thus sustainable PA behavior. Additionally, the identified clusters illustrate a large heterogeneity regarding the interaction between leisure-time PA, body mass index and subjective vitality.

## 1. Introduction

The World Health Organization recommends adults to accumulate at least 150–300 min of moderate-intensity aerobic physical activity (PA) or at least 75–150 min of vigorous-intensity aerobic PA or an equivalent combination of these types of PA throughout the week in order to reach positive health outcomes regarding, for example, cardiovascular diseases, type 2 diabetes, adiposity, cancer or all-cause mortality [1]. Additional amounts of PA may lead to additional health benefits [1,2,3,4]. Furthermore, it is recommended to limit sedentary behavior to achieve health benefits since sedentary behavior has its own negative effect on health over and above the negative effect of insufficient PA [1]. Equally important as the benefits for physical health, PA also has positive short-term and long-term effects on cognitive and mental health [1]. Thus, engaging in sufficient amounts of PA and reducing sedentary behavior offers comprehensive positive effects on both physical and psychological well-being. In this study, it is examined how PA during leisure time and through active commuting (AC) typically interacts with aspects of physical competence and well-being in university students.

In order to reflect physical and psychological well-being in an efficient and still highly reliable and valid manner, subjective vitality, defined as the “conscious experience of possessing energy and aliveness” [5], appears to be a well-suited indicator for two main reasons. First, it is expected that the subjective perception of vitality, as it is defined above, depends on the quality of both physical and mental health aspects [5]. Regarding physical aspects, it has been shown that subjective vitality is positively associated with, for example, perceived body functioning, immunological functioning and a healthier reactivity to stressors of the nervous system [5,6,7,8]. Body mass index (BMI), physical symptoms and physical exhaustion are negatively related to subjective vitality [5,6,9,10]. In terms of mental health aspects, subjective vitality exhibits positive relations with, amongst others, positive affect, regulation of negative emotions, coping with life challenges, life satisfaction and self-esteem [5,6,7,8]. Negative associations have been found with neuroticism, negative affect and emotional exhaustion [5,6]. The second reason that makes subjective vitality an appropriate indicator for physical and psychological well-being is that the aspects subjective vitality directly evolves from mostly form part of the everyday physical and mental experience of a human being, which makes it an easily accessible indicator [5].

In view of the positive relation between PA and well-being, e.g., [1,3], findings about positive associations between PA and vitality lend further support for the suitability of using subjective vitality as an indicator for well-being. In a classic experiment by Thayer [11], university students took part in a walking intervention which asked them to engage in a brisk walk for ten minutes while breathing deeply and swinging their arms freely. It was shown that walking increased participants’ self-reported energy and also decreased their tension for two hours. In a study implementing a 16-week intervention including lunchtime and weekend walks, general subjective vitality of previously physically inactive, sedentary employees increased and their fatigue at work decreased. Changes remained for four months after the end of the intervention [12]. Other studies supported the positive effects of short walks of 15 to 30 min on subjective vitality [13,14,15]. However, as the study by Thøgersen-Ntoumani et al. [12] already suggested, the link between PA and well-being is not limited to short-term effects of acute PA on perceived states of well-being. In addition, associations between regular engagement in PA and subjectively perceived vitality over the course of several weeks have been found. For example, Ellingson, Kuffel, Vack and Cook [16] demonstrated that middle-aged women who were moderately active for at least 150 min per week showed significantly higher values in vitality over four weeks than women not meeting this benchmark.

This is the point where the ambiguity of the relation between PA engagement and well-being needs to be considered. Normally, well-being, or in this case more specifically vitality, is examined as a positive consequence of engagement in PA. Although this direction of effect has been supported in a number of studies, e.g., [12,14], the opposite direction of effect, i.e., higher subjective vitality leading to PA, may also be true. This suggests a potential predicament. On the one hand, it is a promising insight that PA is an activity that can lead to a higher subjective vitality as this may in turn help people to persist in this very activity and eventually lead an active, and thus healthy life [17,18,19]. On the other hand, if people lack a certain level of vitality in the first place, they might not even start to engage in PA. Ryan and Frederick [5] associate subjective vitality with a sense of agency and assume that individuals who perceive high levels of vitality tend more to take up and regularly engage in healthy activities in general. This notion was supported by a study that enrolled participants in a well-being enhancement intervention over eight weeks [20]. Although the intervention did not have a specific focus on PA, not only participants’ subjective vitality increased, but also their PA levels. This indicates an effect of vitality on PA, and thus corroborates the reciprocal relationship between PA and vitality [5,20]. 

However, there are certain circumstances and preconditions that should be given so that vitality may more likely affect PA and vice versa. One step before vitality can potentially lead to elevated and sustained PA levels, one needs to be sufficiently satisfied in the need for competence in order to experience vitality in the first place [5,21]. The underlying assumption is that general feelings of incompetence and also specific activities in which one is not sufficiently competent inevitably lead to effortful behavior, which drains energy and thus finally reduces vitality [18,21]. The relevance of competence was illustrated in the study by Goldbeck et al. [14]. They not only demonstrated the positive effect of a 15 min walk on vitality—an activity in which participants were competent—but also examined whether a strength-exploring interview that aimed to support their perceived competence can also help increase their subjective vitality. Results indicated that participants’ subjective vitality increased after completing a questionnaire and being interviewed with regard to their personal strengths. This finding also points to the necessity of clearly experiencing oneself as being competent so that competence may enable vitality. In a recent study, it was shown that perceived physical ability (PPA) only had a positive effect on well-being in individuals who were regularly physically active and thus had the frequent opportunity to experience their competence in physical tasks [22]. The positive association between perceived competence and subjective vitality has been replicated in a number of cross-sectional [5,18,23,24,25,26] and longitudinal studies [27,28,29,30] with diverse samples of different age groups. The longitudinal study by León and Núñez [30] demonstrated the direction of effect as university students with a higher competence satisfaction at the beginning of a semester exhibited a higher subjective vitality by the end of the semester. This underlines the role of perceived competence as a precondition for the experience of vitality. Furthermore, several diary studies have illustrated the immediateness of the association between daily changes in perceptions of competence and resulting changes in subjective vitality. For samples of university students [31,32] and adults [33] in the context of everyday life as well as for a youth sample in the context of gymnastics training [34], day-to-day within-subject variations in competence satisfaction were associated with individual changes in subjective vitality.

The circumstances that, in turn, facilitate the positive effect of PA on vitality mainly refer to the environment. In their study examining the effect of a walking intervention on vitality, Ryan, Weinstein et al. [13] introduced a variation between walking indoors in a set of underground tunnels or outdoors on a natural tree-lined path along a river. Only the participants who did their 15 min walk outdoors experienced an increase in subjective vitality, whereas walking indoors did not foster participants’ vitality. The walking intervention study by Tyrväinen et al. [15] then tested the obvious assumption that especially outdoor PA has the potential to increase well-being with more detail. Participants went for a 30 min outdoor walk either in a built-up city center, in an urban park or in an urban forest. Subjective vitality only increased in the green environments of the park and forest. In the participants who walked in the city center, vitality even decreased. These findings suggest that PA may more likely increase vitality when it takes place in green and natural environments and, on the other hand, might even lead to reduced vitality when it is done in artificial, urban environments that lack sufficient natural elements [15].

The presented findings suggest that PA should be done in a specific way that may support the experience of subjective vitality, which in turn makes it more likely that the engagement in PA is more sustainable [21]. In order to do so, physical activities that one decides to engage in should imply the experience of competence, e.g., [14,21,33] and should ideally take place in outdoor environments that provide sufficient natural, green elements [13,15]. An activity that may fulfill these conditions is AC. Here, competence is assumed to be satisfied as walking or cycling can be done by everyone who is not affected by any chronic or acute diseases that prohibit these activities. Regular AC should therefore also enable the frequent experience of one’s physical competence. In addition, at least parts of AC to, for instance, school, workplace or leisure-time activities inevitably take place outdoors. Furthermore, in many cases, the time that is spent on reaching a destination via AC is supposed to resemble the duration of the vitality-supportive walking interventions described above, e.g., [13,15]. An advantage with regard to supporting PA is that AC generally represents a promising means to facilitate regular engagement in PA as it incorporates it in activities of daily life that have to be done either way. Correspondingly, using active means of transport (i.e., walking or cycling) for the daily life routine of commuting to school or workplace is considered a good opportunity to establish a healthy level of PA [35,36]. The existing literature supports the assumption that AC to school and workplace may not only increase PA but also reduce body weight and foster well-being in youth and adult populations, e.g., [36,37,38,39,40]. In the study by Ruiz-Ariza et al. [39], AC to high school for more than 15 min per day was associated with higher levels of well-being and happiness in adolescents. Based on data from nine countries, Kleszczewska et al. [40] showed that adolescents who do not engage in AC to school exhibited the largest somatic and psychological health complaints, including depression, nervousness, bad mood and sleep problems. Adjusting for distance to school, results further indicated that complaints were lower in adolescents who walked to school. Students cycling to school had the least health problems. These findings underline that AC might fulfil the conditions described above and may thus be a convenient means to support PA in a sustainable and health-promoting way as it incorporates an increase in vitality.

Contrary to children, adolescents and adults, university students have not been the explicit subject of related research, although they represent a specific risk group in terms of PA and health. University is a key context where lifestyle is consolidated affecting the present and future health of students [41,42]. The transition from high school to university is characterized by a decrease in several domains of PA, including AC [43,44,45,46]. Additionally, the time spent on sedentary behaviors like internet use, sitting while studying or hanging out with friends or family increases and quality of dietary intake decreases [42]. Consequentially, with regard to the benchmark of recommended 150 min of moderate PA per week [47], approximately 50% of university students worldwide are considered insufficiently active [48,49]. Lower PA levels are associated with passive means of transport to university [50]. Passive commuters also exhibit a higher probability of suffering from obesity compared to university students who commute actively [51]. The average weight gain of university students is higher than the one of the general population [52]. These findings strongly suggest a decline in health behavior and physical health status in this group, which could continue to influence their long-term health behavior. In line with the positive association between PA and vitality, e.g., [14,16], mental health of university students also seems to be negatively affected by a decline in PA. Individuals who became insufficiently active during the transition from high school to university, exhibited lower vigor and higher fatigue compared to those who stayed physically active [46].

With regard to the presented findings of previous studies, AC to and from university (ACU) might fulfill several conditions that may help university students lead a more active and healthier life and thus represents a promising option for a population that faces high risks regarding physical inactivity. ACU seems to have the potential to combine a rather conveniently produced substantial level of everyday outdoor PA with the experience of physical competence and subjective vitality, which in turn may finally make their PA behavior more sustainable. To find out whether and in which form this promising pattern of aspects can be found in university students, a person-oriented analysis is conducted in this study. Thus, it refrained from using a traditional variable-based approach, which sets its primary focus on the population as a whole and thus implicitly assumes that any given individuals are legitimate representations of a homogeneous population they form part of [53]. Consequentially, the variable-based approach examines relations of variables on a between-person level across a given population [54], and so may not detect potentially manifold interactions of constructs that may differ substantially between different groups of individuals. Contrarily, the focus of the person-oriented approach lies on the individual instead of on the population as a whole [55]. This approach explicitly builds upon the assumed existence of subgroups within a population that are characterized by specific differences in how the variables of interest relate to each other [53]. Consequentially, the main purpose of person-oriented analyses is to detect several homogeneous groups of individuals, so-called clusters, that exhibit similar patterns regarding their values in the variables [55,56]. Each of the groups then consists of members with homogeneous patterns that are distinct from the patterns of the members of other groups. This allows the person-oriented approach to detect both within-person interactions and between-person differences. Therefore, by use of a person-oriented approach in a sample of university students, this study examines how leisure-time physical activity (LTPA), ACU by walking, ACU by cycling, subjective vitality, PPA (i.e., PA-related competence), and BMI typically co-exist within individuals of different clusters. It is thus analyzed whether university students who actively commute to and from university not only benefit from that in terms of higher PA in a specific domain and healthier BMI, but also exhibit greater awareness of their physical abilities and more favorable subjective well-being, which could ultimately provide a solid basis for regular and sustainable engagement in PA.

## 2. Materials and Methods

### 2.1. Sample

The final sample comprised 484 participants (59.3% women), after excluding 34 participants due to missing data. Age of the participants ranged between 18 and 29 years (M = 21.3 years, SD = 2.5). They were recruited via convenience sampling from two urban universities (Universitat de València, Universitat Politècnica de València) in Valencia, Spain. The university students lived in the city of Valencia and in other municipalities of the Valencian region. Data were assessed by a self-report paper-pencil questionnaire. Completion of the questionnaire required on average about 20 min. Informed consent was obtained from all participants before data collection. The study was conducted in accordance with the Declaration of Helsinki, and an institutional approval was received from the involved universities.

### 2.2. Measures

#### 2.2.1. Leisure-Time Physical Activity

The Spanish version of the Global Physical Activity Questionnaire (GPAQ) [57] was used to measure LTPA. The GPAQ allows assessing the total minutes per week spent with moderate-to-vigorous PA (MVPA) during leisure time. In the following analysis, MVPA is used as an indicator for LTPA, which does not include the time spent on ACU. The Spanish GPAQ is a suitable and acceptable instrument for measuring PA according to a reliability and validity study carried out in an adult population [57]. Moreover, it has already been successfully applied in a sample of university students [44].

#### 2.2.2. Subjective Vitality

The Spanish version [6] of the Subjective Vitality Scale [5] was used. It has six items (e.g., “I have energy and spirit.”). Participants responded by using a 7-point Likert scale ranging from 1 (“not at all true”) to 7 (“very true”). The scale showed a unidimensional factor structure and adequate psychometric properties [6]. In the present study, estimates of internal consistency were acceptable, with Cronbach’s alpha values of 0.84 for men and 0.89 for women.

#### 2.2.3. Perceived Physical Ability

Participants’ self-ratings of their physical activity-related competence were assessed by the perceived physical ability scale [58] translated into Spanish [59]. The ten items (e.g., “Because of my agility, I have been able to do things which many others could not do.”) were rated on a 6-point Likert scale ranging from 1 (“strongly disagree”) to 6 (“strongly agree”). The scale exhibited a unidimensional structure and adequate psychometric properties [58]. In the present study, Cronbach’s alpha was 0.74 for men and 0.78 for women.

#### 2.2.4. Active Commuting to and from University

ACU was assessed with the following item: “How often do you use each of the following options to go to and from university?” [50]. Response options were walking, cycling, bus, car, train/metro/tram and motorbike. Participants indicated the average number of trips per week (to and from university) as well as usual minutes per trip in each mode of commuting. The total minutes per week spent walking and cycling were calculated. This approach has been validated in a previous study [50].

#### 2.2.5. Demographic and Anthropometric Data

Gender, age, socioeconomic status (SES) and BMI were assessed as well. The following question was used to assess SES: “In general, how would you rank your socioeconomic status?”. Participants answered by use of a scale ranging from 1 (“low”) to 5 (“high”) [60]. Body mass index (kg/m^2^) was calculated based on self-reported height and weight.

### 2.3. Data Analysis

MVPA, ACU by walking, ACU by cycling, subjective vitality, PPA and BMI were included as input variables in a self-organizing maps (SOM) analysis [61] in order to identify typical clusters regarding the interplay of these variables of interest. It was conducted with Matlab R2018a (Mathworks Inc., Natick, MA, USA) and the SOM toolbox (version 2.0 beta) for Matlab [62]. The dataset is provided as Appendix A.

The SOM analysis confers several advantages [63]. It is based on an unsupervised algorithm for a potentially non-linear data structure. The statistical power is not affected by the number of included input variables. It identifies clusters with a higher accuracy than regularly used clustering techniques [64,65]. Furthermore, SOM facilitates understanding of the results by presenting them in both numerical and graphic form.

The SOM analysis follows a three steps procedure [66]: It starts with the construction of the neuron network, which consists of 12 × 7 neurons (height × width) for females and 10 × 7 neurons for males in this study. The number of neurons that a network contains is determined by the square root of the number of participants multiplied by five. The second step is the initialization, in which each neuron of the network is assigned a start weight for each input variable in two different ways (i.e., randomized and linear initialization). The training finally represents the third and last step, in which the start weights of the neurons are modified by use of two different training algorithms (i.e., sequential and batch) [67]. The training is an iterative process that modifies the neuronal weights until the best-fitting solution is found. This process is influenced by several factors. Every participant is represented by an input vector, which are introduced to the network. In the next step, each neuron in the network tries to gather the input vectors by adapting its initial weight vector so that they establish the smallest Euclidean distance between its weight vector and the input vector. Accordingly, the weight vector of the winning neuron eventually has the closest value to the input vectors of the participants it has gathered [68]. A total of two aspects determines the magnitude of this adaptation: (a) the learning ratio, which starts with a high value that is subsequently reduced during the training process and (b) the neighborhood function, which determines the adaptation of the winning neuron and the rest of the neurons. More precisely, the adaptation of the winning neuron and its closest neighbors is better than the adaptation of the neurons being further away in the network [66].

To consider the fact that the analysis depends on a random procedure (e.g., initialization and entry order of the input vectors), the process described above was repeated 100 times so that the odds of finding the best-fitting solution are increased. Consequently, 1600 SOMs were produced as two initialization methods, two training algorithms and four neighborhood functions were used (i.e., 100 × 2 × 2 × 4). After multiplying the quantization and topographical errors, the map with the minimum error is chosen [59,68].

In order to create a more ostensive solution, the neurons are finally classified into superordinate clusters according to the values on the input variables with help of a k-means method. To determine the final number of clusters, the quantization error and the number of participants per cluster are compared between the different possible solutions. The quantization error describes the average Euclidean difference between participants’ input vectors and the weight vectors of the clusters they are allocated to [69]. The final cluster solution should exhibit a low quantization error while exhibiting a sufficient number of participants in each cluster in consideration of the total amount of participants (i.e., to avoid clusters with little representation). Based on these superordinate clusters, typical patterns regarding the interplay of the input variables were described for university students.

Separate analyses were conducted for female and male university students because significant gender differences were found in some of the input variables (see Table 1) and in the covariance matrices. The identified clusters were compared in terms of the input variables as well as age using one-factor ANOVAs. The level of significance was set to *p* = 0.05.

## 3. Results

The descriptive statistics for female and male participants are presented in Table 1. The male university students had significantly higher values in BMI, PPA and subjective vitality. Additionally, they reported an average MVPA level that was more than twice as high as the one that the female university students reported.

### 3.1. Females

Figure 1 illustrates the results of the SOM analysis in the female subsample (n = 287).

For each of the six input variables, a neuron network is presented (Figure 1A). Depending on the values in the respective input variable, each participant is placed in a certain neuron (presented as hexagons of different colors). Yellow color represents the highest values of the sample distribution for each variable, dark blue color represents the lowest values. The range of the corresponding numerical values is indicated on the color scale next to each neuron network. Importantly, a given participant is placed in the exact same neuron in each of the six neuron networks. In this way, it becomes apparent which values of the respective variables typically co-exist within a given person.

Figure 1B presents the magnitude of the quantization error depending on the different number of superordinate clusters into which the neurons could be classified in order to present a more ostensive solution. The selection of six clusters appears to be the best-fitting solution as it combines the lowest quantization error with a substantial number of participants per cluster (Figure 1C). Thus, Figure 1D finally presents the six superordinate clusters that describe typical patterns of how the variables of interest may co-exist in female university students.

The six clusters differed significantly in every SOM input variable. Cluster affiliation explained on average 42.9% of participants’ variance in the input variables. Table 2 presents descriptive statistics regarding the input variables as well as demographic and anthropometric data. It also shows the pairwise comparisons between the six female clusters.

Table 3 ranks the mean values in the input variables of the respective clusters according to quintiles based on the distribution of the female subsample.

In the following, the most important characteristics of each of the female clusters are highlighted.

Participants of cluster 1 (named “walkers”) exhibited the highest amount of ACU by walking, while all the other input variables were either average or below average compared to the rest of the female clusters. Cluster 2 (“vital sloths”) had the lowest amounts of MVPA and ACU (walking and cycling taken together). However, they exhibited the lowest BMI and the second highest vitality. Cluster 3 (“feel-good leisure-time actives”) comprised the female university students with the highest MVPA level and the highest values in vitality and PPA. Their ACU was on an average level. Participants of cluster 4 (“unvitals”) exhibited average levels of MVPA and ACU. They had the lowest vitality and the second lowest PPA. Cluster 5 (“perceived incompetents”) was widely similar to cluster 4 with average amounts of MVPA and ACU, as well as the second lowest vitality and the lowest PPA. Their BMI, however, was above average. Cluster 6 (“bikers”) had a relatively high MVPA level and the highest ACU by cycling. Their vitality was above average, and they had the highest BMI of the sample.

### 3.2. Males

Figure 2 illustrates the results of the SOM analysis for the male university students (n = 197).

Considering the quantization error (Figure 2B) and the number of participants per cluster (Figure 2C), a selection of six clusters also appears most appropriate for the male participants.

The clusters exhibited significant differences in every SOM input variable except for BMI. Cluster affiliation explained on average 40.0 % of the variance in the input variables (47.3 % when BMI is excluded). Table 4 shows the descriptive statistics for the input variables as well as demographic and anthropometric data. Pairwise comparisons between the six male clusters are also presented.

Table 5 ranks the mean values in the input variables of the respective clusters according to quintiles based on the distribution of the male subsample.

The most remarkable characteristics of each of the male clusters are described in the following.

Cluster 1 (“cycling leisure-time actives”) comprised participants with the highest MVPA and ACU by cycling, but the lowest ACU by walking. Their vitality and PPA were at an average level compared to the other male clusters. Cluster 2 (“walking leisure-time actives”) had the second highest amounts of MVPA and ACU by walking, the remaining input variables were at an average level. Participants of cluster 3 (“walkers”) exhibited the highest ACU by walking together with an average MVPA. Vitality was below average. Cluster 4 (“feel-good leisure-time actives”) exhibited a high MVPA level and the highest values in vitality and PPA. ACU was at an average level. Cluster 5 (“passive commuters”) had the second lowest amount of MVPA and the lowest amount of ACU (walking and cycling taken together). Vitality and PPA were at an average level. Cluster 6 (“feel-bad sloths”) comprised participants with the lowest values in MVPA, vitality and PPA paired with an average ACU.

## 4. Discussion

In this study, a SOM analysis was conducted to find out whether ACU represents a helpful means for university students to establish a vitality-supportive and thus sustainable PA behavior. The results indicate a large heterogeneity, as for both female and male university students, six clusters were identified that describe how MVPA, ACU, subjective vitality, PPA and BMI may co-occur within a given individual. In the following, the results presented in the figures and tables will be summarized, discussed and interpreted with regard to previous findings.

For both females and males, the neuron networks show that university students who report the lowest subjective vitality and PPA are also the ones who neither exhibit an MVPA level nor an amount of ACU that would be above average. In this way, the female “unvitals” and “perceived incompetents” as well as the male “feel-bad sloths” support the plausibility and the nomological validity of the results with regard to previous research, e.g., [5,12,14,16,20]. 

At the other end of the well-being spectrum, the identification of the female and male “feel-good leisure-time actives” further strengthens the assumption that PA is positively associated with subjective vitality since both clusters report the highest subjective vitality while exhibiting an amount of weekly MVPA that was clearly above average. This relation seems to be clearer in the female subsample compared to the male one as most female clusters with a merely average MVPA level exhibited a subpar subjective vitality. However, there are also some clusters that suggest an ambiguous relation between PA behavior and subjective vitality. The female “vital sloths” represent one of these exceptions since they had the lowest MVPA level but the second highest vitality. It could be speculated whether their high vitality might be due to their strikingly low BMI [9], which could be an indicator of a healthy and thus vitalizing diet that might partly compensate for the low MVPA level [70]. In the male subsample, especially the identification of the “cycling leisure-time actives” is not in line with the assumed positive relation between PA and vitality as they had by far the highest MVPA and ACU by bike and still only reported an average subjective vitality.

The specific relation between ACU by walking and subjective vitality is less ambiguous and does not argue for a positive role of walking to and from university regarding the students’ vitality. When calculated for five days per week, both the female and male “walkers” invested on average almost 25 min on walking to and from university per day. This lies perfectly within the boundaries of the range of durations that participants of vitality-supportive walking interventions had to walk in previous studies, e.g., [12,14,15]. Still, the “walkers” reported subjective vitality levels that were below average. Furthermore, with regard to their merely average LTPA level, or in the case of the female “walkers’” even modest amount of LTPA [1], the results indicate that ACU by walking does not seem to compensate for the lack of a sufficient and thus potentially vitality-supportive LTPA. This is particularly unfortunate for the female “walkers” as ACU by walking seems to be their main opportunity for engaging in PA. 

The role of ACU by bike for the experience of subjective vitality is highly equivocal. Both the female “bikers” and their male counterpart, the “cycling leisure-time actives”, are highly physically active in their free time. This allows us to assume that these students generally like to be physically active and might see ACU by bike as one of many opportunities to engage in PA. This circumstance, however, makes it impossible to discuss the unique role of cycling to university regarding vitality within a cross-sectional design. In addition, whereas female “bikers” exhibited a relatively high vitality, the male “bikers” reported an average level of subjective vitality. Therefore, the relation between ACU by bike and subjective vitality finally remains unclear in this study. 

Taken together, a clear positive relation between ACU and subjective vitality was not found in this study. This finding might be explained when considering the described circumstances and preconditions that are necessary for this relation to be established. Since walking fosters vitality especially when it takes place in a natural environment that contains many green elements [13,15], the surroundings of the universities that students actively commute to might play a role. Although Valencia is a city that offers a high walkability [71], the high residential density and the high temperatures especially during spring and summer lead to a preponderance of built-up areas, sealed surfaces and dry soils that provide little green spaces. These environmental circumstances may harm the potential positive effects of ACU on subjective vitality. Furthermore, the clusters who actively commute to university did not show elevated levels of PPA. This could imply that university students do not think of walking and cycling to university as a domain in which particular physical abilities are needed, which could be explained by the relatively low demands regarding technical skills and intensity. This would preclude ACU from helping university students to experience their abilities [14,22]. Consequentially, these activities would not be suited to strengthen subjective vitality.

On the one hand, this finding indicates the partial independence of subjective vitality from PA or at least from specific types of PA. On the other hand, the thought that university students who actively commute to university did not exhibit elevated levels of PPA and as a potential consequence were not strengthened in their vitality, underlines the relevance of competence regarding subjective vitality [14,21]. These considerations are further supported by taking a closer look at the respective neuron networks and clusters. Especially the identification of the female “vital sloths” and the male “passive commuters” shows that the subjective experience of a relatively high vitality is more likely to occur, and thus also occurs in different individuals, than a relatively high level of LTPA and/or ACU. This illustrates clearly that subjective vitality is partially irrespective of PA behavior (comprising both LTPA and ACU) and rather depends on other factors that were not part of the analysis, with sedentary behavior, sleep or social interaction being possible determinants of subjective vitality [72,73,74,75].

The concordance between subjective vitality and competence seems to be stronger. There are no clusters whose vitality and PPA would not be on a comparable relative level. Especially the “feel-good leisure-time actives” as well as the female “unvitals” and the male “feel-bad sloths” demonstrate that the occurrence of high PPA went along with high subjective vitality, whereas individuals who reported a low PPA also reported a low vitality. This co-occurrence appeared to be even stronger for male than for female university students. Remarkably, this finding underlines the role of competence regarding vitality despite the fact that only one specific aspect of competence, namely the one related to PA, has been addressed in this study. Taken together, it may eventually be assumed that the subjective feeling of vitality is not necessarily related with the actual PA behavior but rather with more theoretical considerations about one’s capability to engage in PA.

Including BMI as a SOM input variable did not lead to a more detailed picture. For the male university students, BMI was not connected to the other variables of interest in a way that would correspond to a specific identifiable pattern. Despite some differences between female clusters in BMI, a clear pattern regarding its intraindividual interaction with the other variables could not be identified in the female subsample either. For example, both the clusters with the lowest BMI (“vital sloths”) and the one with the highest BMI (“bikers”) exhibited a vitality above average. One reason might be the small variance in BMI in the present sample as the mean BMI of every female and male cluster was within the range of normal weight.

The main strength of this study lies in the application of a person-oriented approach. By means of the SOM analysis, the different possibilities how the variables of interest may interact within different groups of individuals are emphasized. This way, between-person differences can be detected while the main focus still lies on the individual [55]. By offering this combination of insights, non-linear relations between the variables of interest become more obvious and tangible. In a variable-based approach, non-linear relations between variables, which are caused by nothing other than the existence of groups of individuals who do not contribute to a supposed predominant linear relation between variables, simply decrease the magnitude of correlations between the variables. Consequentially, even in case of weak correlations, the main conclusions still refer to the association between the variables, which to some extent takes the attention away from the fact that the proclaimed association between variables does not apply to the majority of the examined sample. Contrarily, person-oriented analyses do not even try to find results that would apply to the vast majority of the sample, but rather draw attention to the fact that the interplay of the variables of interest differs between individuals. This advantage is highlighted in this study since the presented findings indicate that, for example, the relation between LTPA and vitality seems to be more subject to individual differences and is thus more fragile than the interplay of perceived PA-related competence and vitality. Another strength is the focus on university students, who represent a neglected population in studies examining health behaviors although previous research has identified university students as a specific risk group regarding physical activity and health, e.g., [48,51,52]. Furthermore, the consideration of behavioral, psychological and anthropometric constructs represents a theoretical, content-related strength.

A limitation of this study is the subjective assessment of LTPA and ACU by means of a self-administered questionnaire. Although the used items and scales are considered reliable and valid instruments [44,50,57], objective devices like accelerometers would offer the possibility of eliminating subjective and retrospective errors in assessing PA behavior [76,77]. Another limitation is the cross-sectional study design, which does not allow for definite conclusions about how the variables of interest affect one another. Furthermore, including more variables in the analysis, such as sedentary behavior, sleep, social interaction or diet would have contributed to a more detailed picture and further explanations of the presented results. Despite the high suitability of subjective vitality as an indicator for subjective well-being, e.g., [5], it would be interesting to examine if the results could be replicated when using other indicators of subjective well-being, such as life satisfaction or positive affect, or indicators of ill-being, such as negative affect. Information regarding the type of LTPA that the participants engage in might have possibly allowed for more detailed explanations for the identification of individuals with a high PA level but only average subjective vitality. Additionally, it would have been interesting to track the routes that the students take on their way to university when engaging in ACU. With this information, it could be examined whether the negligible role of ACU for the students’ vitality can be explained by the type of environment that the students face while commuting to university [13,15]. In addition, conducting interviews with the students about their perception of their physical abilities when actively commuting would further contribute to an informed interpretation of the results concerning the role of PPA for vitality in the context of ACU [5,22]. Lastly, the potential effects of seasonality on ACU and on its interplay with the other variables of interest could not be examined since the seasonal changes in weather conditions in the Valencian region are not large enough to considerably affect ACU [78,79].

## 5. Conclusions

The findings of this study do not advocate ACU as a promising means that could lead university students to experience their physical abilities, support their subjective well-being and ultimately provide a foundation for sustainable engagement in PA. The association between university students’ PA behavior and their subjective vitality is mainly subject to individual differences. A linear relation connecting these variables must therefore be questioned with regard to the identified clusters. Students’ subjective experience of vitality does not seem to depend on the level of ACU. This could be explained by the widespread barren environment that the students who participated in this study face during ACU, since it contains few green elements and thus rather drains energy. Another reason might be that ACU does not lead university students to experience their physical ability, which therefore does not support vitality. Generally, however, self-ratings of physical ability appear to go along with the experience of subjective vitality, which suggests that university students’ potential ability to be physically active might be more important for subjective vitality than their actual engagement in PA. The role of BMI remains unclear.

## Figures and Tables

**Figure 1 ijerph-19-07239-f001:**
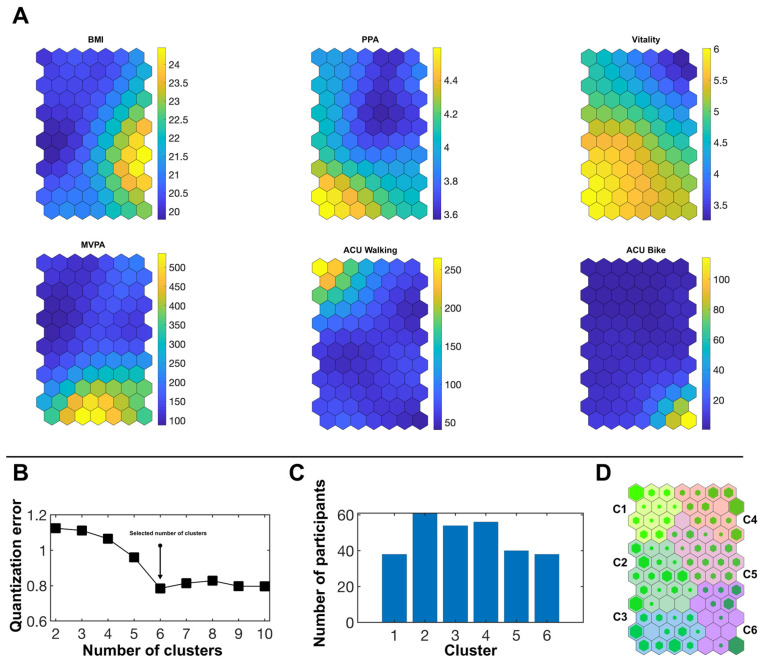
Neuron networks, superordinate clusters and hits obtained by the SOM analysis in the female subsample. (**A**) Neuron networks for the SOM input variables. Yellow color represents the highest values of the sample distribution for each variable, dark blue color represents the lowest values. (**B**) Quantization error according to the possible number of clusters selected. (**C**) Number of participants per cluster for the chosen cluster solution. (**D**) Hits map with the six superordinate clusters C1–C6. The more a neuron is filled with green, the higher the number of participants assigned to the neuron. BMI = body mass index, PPA = perceived physical ability, MVPA = moderate-to-vigorous physical activity, ACU = active commuting to and from university.

**Figure 2 ijerph-19-07239-f002:**
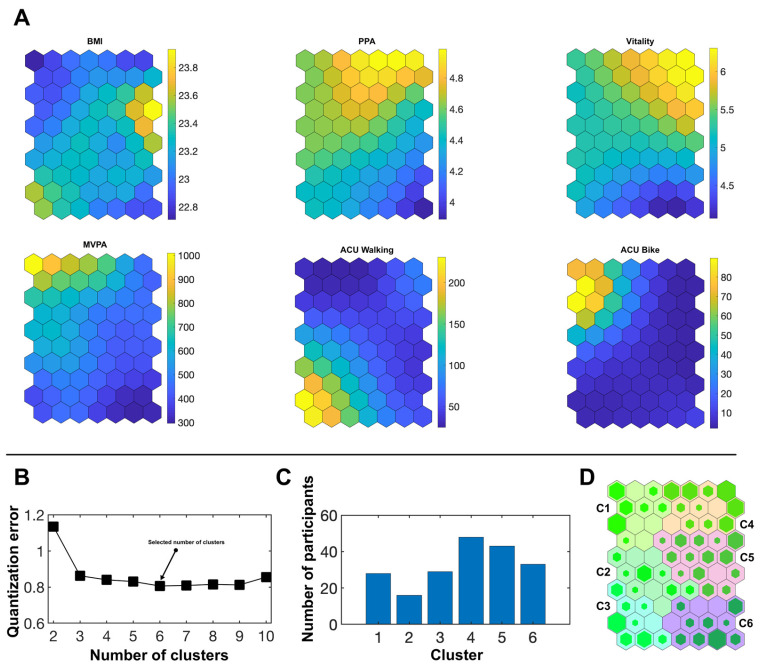
Neuron networks, superordinate clusters and hits obtained by the SOM analysis in the male subsample. (**A**) Neuron networks for the SOM input variables. Yellow color represents the highest values of the sample distribution for each variable, dark blue color represents the lowest values. (**B**) Quantization error according to the possible number of clusters selected. (**C**) Number of participants per cluster for the chosen cluster solution. (**D**) Hits map with the six superordinate clusters C1–C6. The more a neuron is filled with green, the higher the number of participants assigned to the neuron. BMI = body mass index, PPA = perceived physical ability, MVPA = moderate-to-vigorous physical activity, ACU = active commuting to and from university.

**Table 1 ijerph-19-07239-t001:** Descriptive statistics and pairwise comparisons between genders.

	Weight Status (% Normal/Overweight)	SES	Age (Years)	BMI (kg/m^2^)	PPA	Subjective Vitality	ACU Walking (min/Week)	ACU Bike (min/Week)	MVPA (min/Week)
Men	83.2/16.8	3.09	21.54	23.26	4.51	5.32	79.25	18.02	520.25
(n = 197)		(0.66)	(2.29)	(2.43)	(0.53)	(0.97)	(88.27)	(46.30)	(400.94)
Women	92.7/7.3	3.07	21.11	21.34	4.01	4.90	85.55	14.11	217.06
(n = 287)		(0.67)	(2.61)	(2.68) ^a^	(0.53) ^a^	(1.04) ^a^	(88.34)	(45.88)	(283.93) ^a^
Total	88.8/11.2	3.08	21.29	22.12	4.21	5.07	82.99	15.70	340.46
(n = 484)		(0.66)	(2.49)	(2.75)	(0.59)	(1.03)	(88.27)	(46.04)	(367.70)

Note. SES = socioeconomic status; BMI = body mass index; PPA = perceived physical ability; ACU = active commuting to and from university; MVPA = moderate-to-vigorous physical activity; ^a^ indicates significant differences between genders (*p* < 0.001).

**Table 2 ijerph-19-07239-t002:** Pairwise comparisons between women’s clusters.

	Weight Status (% Normal/Overweight)	SES	Age (Years)	BMI (kg/m^2^)	PPA	Subjective Vitality	ACU Walking (min/Week)	ACU Bike (min/Week)	MVPA (min/Week)
Cluster 1	100/0	3.05	20.97	20.40	4.04	4.66	230.52	0.00	117.11
(n = 38)		(0.40)	(2.81)	(1.77) ^5,6^	(0.38) ^3,4,5^	(0.60) ^2,3,4,6^	(101.73) ^all^	(0.00) ^6^	(144.67) ^3,6^
Cluster 2	100/0	3.02	20.52	19.57	4.00	5.58	51.13	5.43	94.75
(n = 61)		(0.65)	(2.59) ^3^	(1.36) ^3,4,5,6^	(0.40) ^3,4,5^	(0.52) ^1,3,4,5^	(44.36) ^1^	(17.07) ^6^	(114.14) ^3,6^
Cluster 3	98.1/1.9	3.26	21.56	20.89	4.61	5.84	71.70	6.44	505.65
(n = 54)		(0.62) ^4,5^	(2.65) ^2^	(1.68) ^2,5,6^	(0.41) ^all^	(0.56) ^all^	(70.97) ^1^	(20.77) ^6^	(353.99) ^all^
Cluster 4	98.2/1.8	2.98	21.16	21.18	3.77	3.35	66.02	3.75	155.36
(n = 56)		(0.67) ^3^	(2.94)	(2.21) ^2,5,6^	(0.46) ^all^	(0.56) ^all^	(60.30) ^1^	(17.22) ^6^	(269.63) ^3,6^
Cluster 5	90.0/10.0	2.90	21.08	22.56	3.44	4.58	72.20	0.00	131.63
(n = 40)		(0.63) ^3^	(2.42)	(2.75) ^all^	(0.37) ^all^	(0.49) ^2,3,4,6^	(54.15) ^1^	(0.00) ^6^	(182.95) ^3,6^
Cluster 6	60.5/39.5	3.18	21.53	24.71	4.06	5.36	58.34	83.16	284.08
(n = 38)		(0.93)	(1.98)	(3.23) ^all^	(0.32)^3,4,5^	(0.59) ^1,3,4,5^	(75.68) ^1^	(95.13) ^all^	(288.97) ^all^
Total	92.7/7.3	3.07	21.11	21.34	4.01	4.90	85.55	14.11	217.06
(n = 287)		(0.67)	(2.61)	(2.68)	(0.53)	(1.04)	(88.34)	(45.88)	(283.93)

Note. SES = socioeconomic status; BMI = body mass index; PPA = perceived physical ability; ACU = active commuting to and from university; MVPA = moderate-to-vigorous physical activity; n = number of participants in the cluster. Data are expressed as mean (standard deviation of the mean); superscript numbers indicate significant differences to the respective cluster(s) (*p* < 0.05); ^all^ = significant differences to all the other clusters (*p* < 0.05).

**Table 3 ijerph-19-07239-t003:** Female clusters’ mean values in the input variables according to quintiles.

	BMI	PPA	Subjective Vitality	ACU Walking	ACU Bike	MVPA
Cluster 1 “walkers”	**-**	**o**	**-**	**++**	**o**	**o**
Cluster 2 “vital sloths”	**-**	**o**	**+**	**o**	**o**	**o**
Cluster 3 “feel-good leisure-time actives”	**o**	**++**	**++**	**o**	**o**	**++**
Cluster 4 “unvitals”	**o**	**-**	**- -**	**o**	**o**	**o**
Cluster 5 “perceived incompetents”	**+**	**- -**	**-**	**o**	**o**	**o**
Cluster 6 “bikers”	**++**	**o**	**+**	**o**	**+**	**+**

Note. BMI = body mass index; PPA = perceived physical ability; ACU = active commuting to and from university; MVPA = moderate-to-vigorous physical activity. Quintiles are indicated as follows: 1st quintile = - -, 2nd quintile = -, 3rd quintile = o, 4th quintile = +, 5th quintile = ++. The actual value of the 5th quintile in ACU by bike is 0 since less than a fifth of the female subsample commutes to university by bike (11.5%). Cluster 6 is still ranked with a + because around half of the students in cluster 6 (47.4%) engage in ACU by bike.

**Table 4 ijerph-19-07239-t004:** Pairwise comparisons between men’s clusters.

	Weight Status (% Normal/Overweight)	SES	Age (Years)	BMI (kg/m^2^)	PPA	Subjective Vitality	ACU Walking (min/Week)	ACU Bike (min/Week)	MVPA (min/Week)
Cluster 1	92.9/7.1	3.21	22.00	22.74	4.59	5.36	15.93	102.71	822.14
(n = 28)		(0.63) ^3^	(2.64) ^6^	(1.72)	(0.38) ^4,6^	(0.77) ^4,6^	(27.38) ^2,3,4,6^	(74.31) ^all^	(602.02) ^3,4,5,6^
Cluster 2	81.2/18.8	3.06	20.81	23.46	4.55	5.25	102.37	6.25	674.69
(n = 16)		(0.57)	(1.33) ^4^	(2.12)	(0.33) ^4,6^	(0.34) ^4,6^	(26.07) ^all^	(17.46) ^1^	(315.91) ^3,5,6^
Cluster 3	72.4/27.6	2.83	20.97	23.66	4.44	5.04	239.31	0.69	433.28
(n = 29)		(0.76) ^1,5^	(2.11) ^4^	(2.28)	(0.41) ^4,6^	(0.77) ^4,5,6^	(87.71) ^all^	(3.71) ^1^	(324.74) ^1,2,4^
Cluster 4	91.7/8.3	3.04	22.15	23.10	4.96	6.22	63.19	1.98	627.60
(n = 48)		(0.74)	(2.27) ^2,3,6^	(1.94)	(0.40) ^all^	(0.53) ^all^	(56.30) ^1,2,3,5^	(8.43) ^1^	(383.54) ^1,3,5,6^
Cluster 5	76.7/23.3	3.28	21.77	23.89	4.46	5.53	34.28	2.98	393.72
(n = 43)		(0.63) ^3^	(2.17)	(3.41) ^6^	(0.38) ^4,6^	(0.50) ^3,4,6^	(34.31) ^2,3,4,6^	(11.17) ^1^	(222.59) ^1,2,4^
Cluster 6	81.8/18.2	3.06	20.82	22.67	3.91	3.96	63.09	10.00	274.39
(n = 33)		(0.50)	(2.43) ^1,4^	(2.18) ^5^	(0.55) ^all^	(0.86) ^all^	(58.91) ^1,2,3,5^	(25.98) ^1^	(242.94) ^1,2,4^
Total	83.2/16.8	2.11	21.54	23.26	4.51	5.32	79.25	18.02	520.25
(n = 197)		(0.57)	(2.29)	(2.43)	(0.53)	(0.97)	(88.27)	(46.30)	(400.94)

Note. SES = socioeconomic status; BMI = body mass index; PPA = perceived physical ability; ACU = active commuting to and from university; MVPA = moderate-to-vigorous physical activity; n = number of participants in the cluster. Data are expressed as mean (standard deviation of the mean); superscript numbers indicate significant differences to the respective cluster(s) (*p* < 0.05); ^all^ = significant differences to all the other clusters (*p* < 0.05).

**Table 5 ijerph-19-07239-t005:** Male clusters’ mean values in the input variables according to quintiles.

	BMI	PPA	Subjective Vitality	ACU Walking	ACU Bike	MVPA
Cluster 1 “cycling leisure-time actives”	**o**	**o**	**o**	**-**	**++**	**++**
Cluster 2 “walking leisure-time actives”	**o**	**o**	**o**	**+**	**o**	**+**
Cluster 3 “walkers”	**+**	**o**	**-**	**++**	**o**	**o**
Cluster 4 “feel-good leisure-time actives”	**o**	**++**	**++**	**o**	**o**	**+**
Cluster 5 “passive commuters”	**+**	**o**	**o**	**-**	**o**	**o**
Cluster 6 “feel-bad sloths”	**o**	**- -**	**- -**	**o**	**o**	**-**

Note. BMI = body mass index; PPA = perceived physical ability; ACU = active commuting to and from university; MVPA = moderate-to-vigorous physical activity. Quintiles are indicated as follows: 1st quintile = - -, 2nd quintile = -, 3rd quintile = o, 4th quintile = +, 5th quintile = ++. The actual value of the 5th quintile in ACU by bike is 0 since less than a fifth of the male subsample commutes to university by bike (19.8%). Cluster 1 is still ranked with a ++ because the vast majority of the students in cluster 1 (85.7%) engage in ACU by bike.

## Data Availability

The data presented in this study is available online as Appendix A.

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
