# Peer review of "Do Active Commuters Feel More Competent and Vital? A Self-Organizing Maps Analysis in University Students"

_ijerph, 2022, doi:10.3390/ijerph19127239_

Round 1

Reviewer 1 Report

The authors describe a study in which associations between leisure-time activity, active commuting to and from university by walking and cycling, subjective vitality and physical-activity related competence were investigated in students from Valencia (Spain). They run a self-organizing maps analysis to report results.

The manuscript can be recommended for publication almost as is. Congratulations!

I only have two small change requests for the manuscript:

The authors do not report how they arrived at the number of neurons for the SOM analysis for the female and male students. A brief explanation of this would be helpful.

Table 1, Table 2, and Table 4: p-values are shown in the footnote. These should be displayed without a 0 before the decimal point.

Reviewer 2 Report

Conclusions could be a bit more extensive for discussing another subjective well-being variables that could be related to ACU and could be included in further studies.  Might instruments other than the Subjective Vitality Scale lead to different results or associations? 

In Conclusions, I find this sentence not totally clear: "This could be explained by the dry environment that the students face during ACU, which rather drains energy".  Dry environment as the opposite of humid environment?  Any seasonal changes to consider here?

Is there information that is useful to clarify how do seasonal changes reflect in ACU behaviour and how would it affect the results?

Were the response samples obtained by face-to-face interviews or web surveys or any other methods?

Reviewer 3 Report

The title is clear and describes the content of the manuscript.

The summary also complies with the corresponding sections.

In the introduction and justification, the background of the problem is clearly stated, as well as those issues that are not clear or require resolution.

Regarding the objectives, it is recommended to formulate secondary objectives so that they describe in a more detailed way the variables that are going to be analyzed.

The results are well structured, synthesizing extensive information that is well organized.

On the other hand, the initials (PA, AC, AC,…) make it very difficult to read them in the Discussion and Conclusions section.

Line 451, avoid including “figures 1 and 2”.

Lines 587 to 589. “This could be explained by 587 the dry environment that the students face during ACU, which rather drains energy. An-588 other reason might be that ACU does not lead university students to experience their 589 physical ability, which therefore does not support vitality”. It is recommended to put the interpretation of the results in the Discussion section.

It is also recommended to expand the conclusions, connecting them with the objectives and the main results.
